# Narrow genetic diversity in germplasm from the Guinean and Sudano-Guinean zones in Benin indicates the need to broaden the genetic base of sweet fig banana (*Musa acuminata* cv Sotoumon)

**Dènoumi B. E. Capo-Chichi**[ID]*, **Dèdéou A. Tchokponhoué**[ID], **Dêêdi E. O. Sogbohossou**, **Enoch G. Achigan-Dako***

Genetics, Biotechnology and Seed Science Unit (GBioS), Laboratory of Crop Production, Physiology and Plant Breeding (PAGEV), Faculty of Agricultural Sciences (FSA), University of Abomey-Calavi, Abomey-Calavi, Republic of Benin

* denoumib@gmail.com (DBECC); e.adako@gmail.com (EGAD)

## Abstract

Sweet fig (*M. acuminata* cv. Sotoumon) is an economically important dessert banana in Benin, with high nutritional, medicinal, and cultural values. Nevertheless, its productivity and yield are threatened by biotic and abiotic stresses. Relevant knowledge of the genetic diversity of this economically important crop is essential for germplasm conservation and the development of breeding programs. However, very little is known about the genetic makeup of this cultivar in Benin. To advance the understanding of genetic diversity in sweet fig banana germplasm, a Genotype-By-Sequencing (GBS) was performed on a panel of 273 accessions collected in different phytogeographical zones of Benin. GBS generated 8,457 quality SNPs, of which 1992 were used for analysis after filtering. The results revealed a low diversity in the studied germplasm (He = 0.0162). Genetic differentiation was overall very low in the collection as suggested by the negative differentiation index (Fstg = -0.003). The Analysis of Molecular Variance (AMOVA) indicated that the variation between accessions within populations accounted for 83.8% of the total variation observed (P < 0.001). The analysis of population structure and neighbor-joining tree partitioned the germplasm into three clusters out of which a predominant major one contained 98.1% of all accessions. These findings demonstrate that current sweet fig banana genotypes shared a common genetic background, which made them vulnerable to biotic and abiotic stress. Therefore, broadening the genetic base of the crop while maintaining its quality attributes and improving yield performance is of paramount importance. Moreover, the large genetic group constitutes an asset for future genomic selection studies in the crop and can guide the profiling of its conservation strategies.

**Data Availability Statement:** All relevant data are within the paper and its Supporting Information files.

**Funding:** The author(s) received no specific funding for this work.

**Competing interests:** The authors have declared that no competing interests exist.

## Introduction

Bananas (*Musa* spp.) are fruit crops of high socio-economic importance as they represent sources of food and income generation for millions of people in developing countries [1, 2]. Originating from Indo-Malesia before spreading to all tropical and subtropical regions [3–5], bananas are giant perennial and monocotyledonous herbs and belong to the Section of *Eumusa* of *Musa* genus. This section contains the major ancestors of edible bananas: *Musa acuminata* Colla (AA) and *Musa balbisiana* Colla (BB) [6, 7], which are wild diploid seed-producing species. *Musa acuminata* is partitioned into several subspecies, among which at least four have been admitted as contributors to the cultivated banana varieties: *banksii, zebrina, burmannica, and malaccensis*. No subspecies have been determined so far in *M. balbisiana* [8–10].

Among the most economically important cultivated bananas in Benin, the sweet fig banana, known as "Sotoumon" (in the Fon local language of the Republic of Benin), is more common and widely grown in almost all municipalities of the country [11]. Sweet fig bananas belong to *M. acuminata* (AA) and the Sucrier subgroup [6, 12]. Sweet fig bananas are more appreciated for their exceptional taste quality than for their agronomic performance, which is considered poor [13].

Unfortunately, sweet fig banana productivity in Benin is limited by biotic and abiotic constraints [11, 14, 15]. Therefore, it is essential to develop high-yielding commercial varieties that meet users' expectations and are adapted to stress conditions. However, developing effective breeding programs requires knowledge of the genetic diversity of cultivars [16, 17]. Previous work undertaken on the genetic diversity of dessert bananas and plantains in Benin using SNP markers revealed high molecular variability depending on their genomic membership [18]. The analysis of genetic diversity within dessert banana, *M. acuminata* cv. AA showed either high genetic diversity [19] or low nucleotide diversity, and high genetic similarity [20, 21]. Furthermore, most of the cultivated AA banana have a hybrid status [22] due to the intraspecific hybridization occurring between different subspecies of *M. acuminata* [8, 23]. Elucidating the genetic diversity within *Musa acuminata* subspecies and subgroups is important to provide valuable information for further breeding plans and conservation activities.

The emergence of molecular markers provides higher accuracy and efficiency for analyzing genetic diversity compared to the phenotypic approach [24, 25]. Several genetic diversity evaluations in *Musa* species were done using different types of molecular markers and technologies such as Amplified Fragment Length Polymorphism (AFLP) [26], Random Amplified Polymorphism DNA (RAPD) [27], Restriction Fragment Length polymorphism [28], Inter Simplified Sequence Repeats (ISSR) [29, 30], Simple Sequence Repeats (SSR) [31, 32], sequence-related amplified polymorphism [33, 34] and Diversity Arrays Technology-sequencing (DArTseq)-GBS [35]. DArTseq enabled a higher SNPs discovery from many accessions for genetic analysis compared to AFLP, RAPD, ISSR, and SSR markers [36, 37]. Among New Generation Sequencing (NGS) technologies, GBS is the simplest, most high-performance, and cost-effective technique [38, 39], widely employed in genetic diversity, populations structures, and phylogeny studies [40, 41].

To our knowledge, the genetic diversity and population structure of sweet fig banana have not been evaluated.

In this study, we explored the genetic variation of 273 sweet fig banana genotypes from Benin to generate information relevant for improving the cultivar and the development of conservation strategies. We hypothesized that: (i) sweet fig banana germplasm encompasses a moderate genetic diversity; (ii) sweet fig banana populations are genetically structured according to their geographic occurrence in Benin.

## Materials and methods

### Plant material collection and description

A total of 273 accessions of sweet fig was collected under the owners' informed consent from March to July 2019 in the Guineo-Congolia and the Sudano-Guinean regions of Benin. The Guineo-Congolia zone has a sub-equatorial climate. Rainfall is bimodal, ranging from 900 to 1400 mm per year. The vegetation includes thickets, mangroves, and semi-deciduous forests. The Sudano-Guinean zone is a transitional area between the sub-equatorial and Sudano-Guinean climates. This zone is characterized by the merging of the two rainfall peaks. The average rainfall varies from 1200 to 1300 mm. The vegetation consists of various types of forests (open, dense, dry, dense humid semi-deciduous, galleries), savannahs with trees and shrubs [42]. Accessions were collected from eight agro-ecological areas, resulting in the different populations. Each population is designated by a code corresponding to the first letters of the districts of origin (Fig 1, S1 Table).

In the Guineo-Congolian zone, we prospected six agro-ecological areas: (i) the sandy coastal plain of the District of Grand-Popo (GPO); (ii) the low river and lake valleys (Mono, Couffo, Ahémé) with alluvial formations in the Districts of Lokossa and Athiémè (LOA) in the Department of Mono; (iii) the Akplahoué ferralitic soil plateau notched by depressions with vertisols in the Districts of Dogbo and Lalo (DOL) in the Department of Couffo; (iv) the Allada ferralitic soil plateau, interspersed with depressions, in the Districts of Zè, Tori-Bossito, and Allada (ZTA) in the Department of Atlantic; (v) the Sakété ferralitic soil plateau and the lowland hydromorphic soils of the Districts of Sakété and Adjohoun (SAD) in the Departments of Ouémé and Plateau respectively; and (vi) the Abomey ferralitic soil plateau and the hydromorphic soils of the Districts of Zogbodomey and Ouinhi (ZOO) in the Department of Zou. The Akplahoué, Allada, and Sakété plateaus follow each other, from west to east. The Akplahoué plateau is separated from the Allada plateau by the Ahémé Lake, the Couffo River, the Aho Channel, and the Tchi Depression. Lake Nokoué, the Sô River, and the Ouémé River separate the Allada Plateau from the Sakété Plateau. The plateaus of Akplahoué, Allada, and Sakété are separated from the Abomey plateau by the Lama Depression [44]. Two agro-ecological areas were surveyed in the Sudano-Guinean zone; these include: (vii) the Central-Western Benin, with Bassila, Bantè and Glazoué (BBG) Districts and (viii) the Central-Eastern Benin, which covers the District of Parakou, Tchaourou and Savè (PST). The soils of those areas mainly belong to the tropical ferruginous type. The Ouémé River and its two main tributaries, the Zou and the Okpara, separate the two populations. Overall, collected accessions belonged to those eight populations (Fig 1) spread across 46 villages of 18 major banana producing municipalities in Benin. Most of the accessions (207) were collected in the Guineo-congolian zone. In this study, we defined a population of Sotoumon as a group of accessions randomly distributed across various production systems within the same agroecological environment as described above. Samples were collected randomly in each population at intervals of 1 km. Two adjacent populations of "Sotoumon" are separated by a geographical distance from 20 to 30 km (e.g. BBG and PST, LAO and DOL). Two distant populations were separated by a distance of 60 to 450 km (e.g. GPO and LOA, SAD and GPO, ZOO and BBG, ZOO and PST, GPO and PST). Based on the International Plant Genetic Resources Institute (IPGRI) Musa descriptors [45] and the Musalogue [12] we classified the accessions of sweet fig banana collected.

### National and local approval

The first author received the approval of the Academic Committee of the Faculty of Agronomic Sciences (FSA), University of Abomey-Calavi (UAC), Abomey-Calavi, Republic of

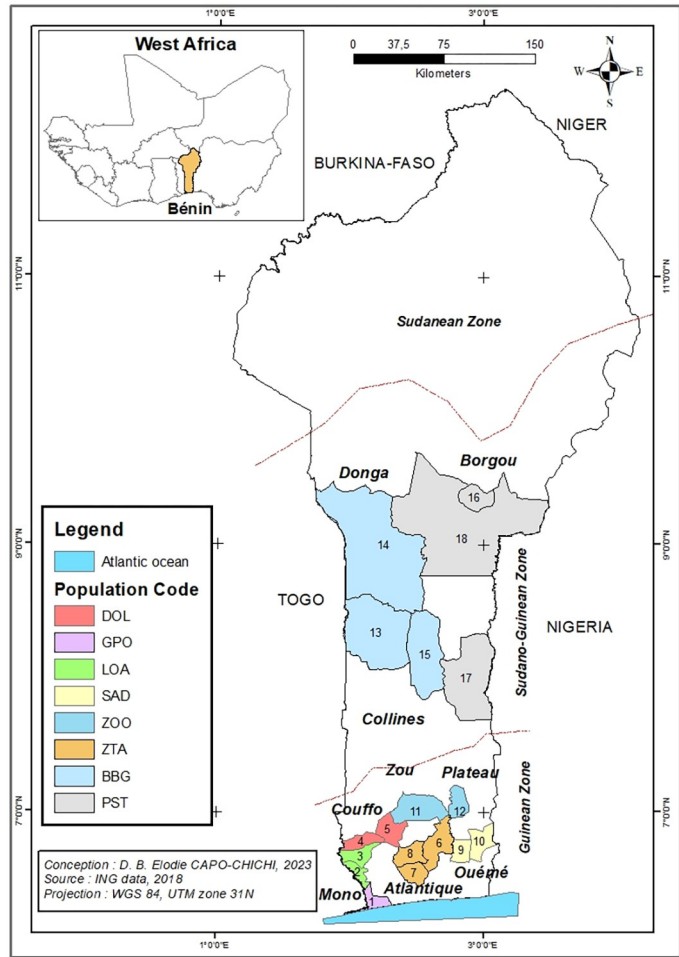

**Fig 1. Map illustrating sweet fig banana accessions sampling points in Benin.** Population codes and associated Districts: GPO: 1. Grand-Popo; LOA: 2. Lokossa, 3. Athiémè; DOL: 4. Dogbo, 5. Lalo; ZTA: 6. Zè, 7. Tori-Bossito, 8. Allada; SAD: 9. Adjohoun, 10. Sakété; ZOO: 11. Zogbodomey, 12. Ouinhi; BBG: 13. Bantè; 14. Bassila, 15. Glazoué; PST: 16. Parakou; 17. Savè, 18. Tchaourou. The base map was obtained from the National Geographic Institute of Benin. The current version of the map was generated using ArcGIS (Version 10.4) [43].

Benin. The plant materials were collected following the National and International Code of Conduct for plant germplasm collection. Before collection, the objective and methodology of the study were presented to the local authorities and sweet fig banana plant owners of each investigated village. We requested and obtained the informed verbal consent of all plant owners to collect the seedlings. All collected plant materials are conserved at the Genetics, Biotechnology and Seed Sciences Unit (GBioS), Laboratory of Crop Production, Physiology, and Plant Breeding, Faculty of Agronomic Sciences, University of Abomey-Calavi (Republic of Benin).

## Genotyping-by-sequencing and SNP quality filtering

About 5–10 g of young leaf samples (three-week-old) were taken from a single plant of each of the 273 sweet fig banana accessions grown in an experimental field in Sékou (Southern Benin). The leave samples were dried in an oven at 35˚C overnight and sent to SEQART AFRICA, (Nairobi, Kenya) for genotyping using Diversity Arrays Technology (DArT) markers. The DNA of each accession was isolated and purified using the Nucleomag Kit to extract plant

genomic DNA. The quality and quantity of DNA were confirmed through 0.8% agarose gel electrophoresis. To prepare the libraries, 100 ng of the DNA from each genotype was used. The library was constructed following the process described by Kilian et al. [46]. The genomic libraries were sequenced using Single Read sequencing runs for 77 bases with the Illumina HiSeq 2500 sequencer. SNP markers were aligned with the reference genome of *Musa acuminata Colla malaccensis*, DH-Pahang V2 [47, 48].

DArTseq markers scoring was achieved using the DArT Proprietary Limited (PL'S) proprietary SNP and SilicoDArT calling algorithms (DArTsoft14). The SNP markers were scored as "0" for homozygous reference allele, "1" for homozygous SNP allele and "2" for heterozygous allele.

The SNPs were filtered using the R package "dartR". SNP markers with a minor allele frequency (MAF) > 0.05, a call rate > 75% and a reproducibility rate of 95% were retained for downstream analyses. Also, only the SNPs with < 20% missing data were retained. After filtering, missing data were imputed using the DArT's KD Compute Optimal Imputation (https://kdcompute.squart-africa.org/kdcompute/login) and SNPs with polymorphic information content (PIC) < 0.05 were removed. Finally, 1992 SNPs (S2 Table) were retained for further analyses.

## Genetic diversity analysis

The genetic diversity of the germplasm was characterized by computing the overall heterozygosity (Ho), expected heterozygosity (He), total gene diversity (Ht) and inbreeding coefficient (Fis). To assess the extent of difference within and among the eight populations, Ho, He, Fis and allelic richness (Rs) were computed per population. These indices were all computed using the dartR package [49].

## Populations differentiation and structure analysis

Overall differentiation index (Fstg) for the germplasm was assessed using the function *stamppFst ()* of the StAMPP package while the pairwise genetic differentiation indices (Fstp) among the eight populations were computed using the function pairwise *WCfst ()* of the hierfstat package. This fixation index allows to highlight the degree of genetic differentiation between two populations [50]. A standard scale of Fst values developed by Wright [51] was used with Fst < 0.05 = low genetic differentiation; Fst = 0.05–0.15 = moderate genetic difference; Fst = 0.15–0.25 = high genetic difference; Fst > 0.25 = very high genetic difference. Population structure was assessed using two approaches. First, an analysis of molecular variance was conducted, using the function *poppr.amova ()* of the Poppr R package [52], to explore the relative contribution of the occurrence region, the population, and the accession to the observed variation. The statistical significance of each source of variation was tested through a bootstrapping based on 999 permutations with the function *randtest ()* of the Poppr package [52]. Second, population structure was approached by implementing a Bayesian clustering model in Structure V.2.3.4. software [53], with a burn-in time and a Markov Chain Monte Carlo (MCMC) iterations set to 10000. Three runs were performed for each population with K varying in the range of 3 to 10. The optimal number of genetic groups was determined using the deltaK method in Structure Harvester software [54, 55].

## Accessions clustering

A neighbor-joining tree was constructed using the Maximum Composite Likelihood model to determine the relationships among accessions studied. This analysis was implemented in

MEGA X [56] with 1000 bootstrap replicates and the result visualized using the Interactive Tree of Life (Version 6) [57].

## Results

### Phenotypic description of sweet fig banana germplasm

Two phenotypes of sweet fig bananas were identified during the in-situ morphological characterization of adult plants at the study locations before sampling. The most common or dominant sweet fig banana phenotype has a light green pseudo stem with brown large blotches. The peduncle is downy (very hairy with short hair). The bunch has a truncated cone and a slightly compact shape with a slightly angled position (Fig 2A). The fruits are curved, with a bright yellow peel color at maturity. The pulp has a cream color at maturity. The rachis forms a curve and is bare.

The male bud has a lanceolate shape (Fig 3A) and may degenerate or persist at bunch maturity. The young bracts overlap at the apex of the male bud. The bract revolt before falling. The external face of the bract has a purple-brown color.

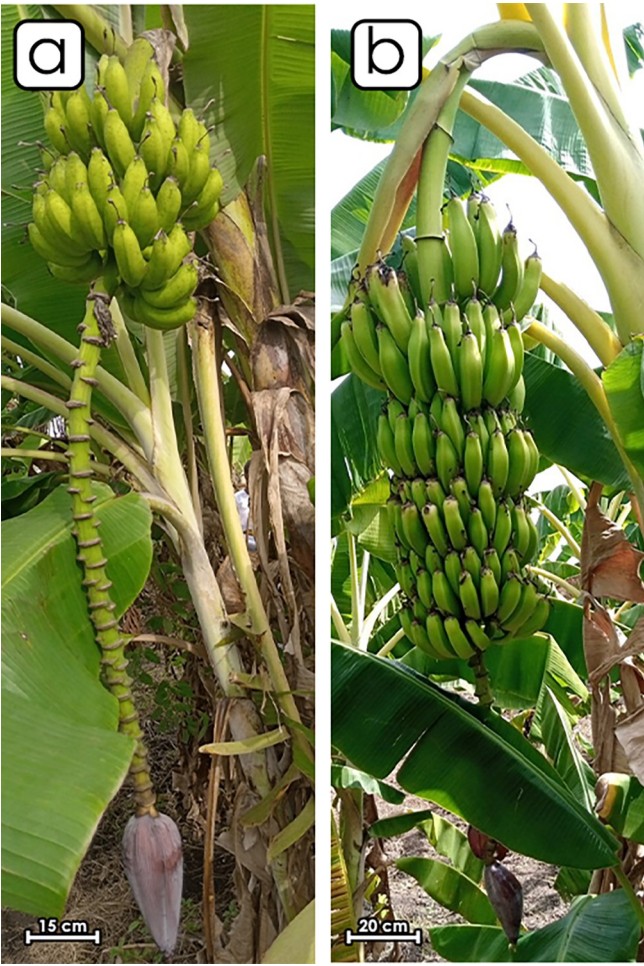

**Fig 2. Morphology of the two phenotypes of sweet fig banana.** (a) Clone set of the major phenotype. (b) Clone set of the minor phenotype.

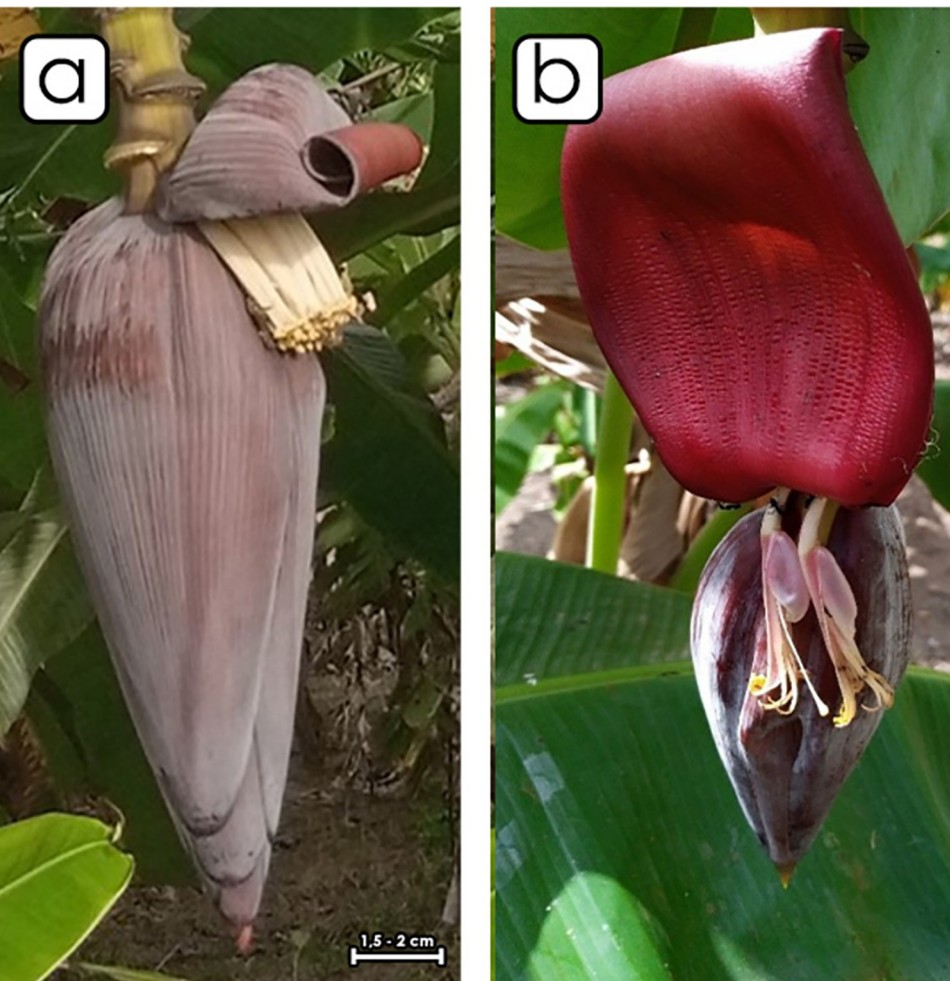

**Fig 3. Morphology of the male bud in the two phenotypes of sweet fig banana.** (a) Major phenotype. (b) Minor phenotype.

The basic color of the male flower compound tepal is cream without pigmentation (Fig 4). The lobe of the compound tepal is Yellow, and the stigma are pink-purple. The basic color of the ovary is cream with no visible sign of pigmentation.

The second phenotype has green-yellow pseudo stem with brown small or light blotches (Fig 2B). The peduncle is hairless. The bunch is cylindrical, compact and handed vertically. The fruits are curved and big, with yellow peel color at maturity. The pulp has light yellow color at maturity. The rachis falls vertically and is bare. The male bud is lanceolate and may degenerate or persist at maturity. The bract external face has a red-purple color. The young bracts overlap at the apex of the male bud. The bract rolls before falling (Fig 3B). The basic color of the male flower compound tepal is pink-purple without pigmentation. The lobe of the compound tepal is bright yellow with cream stigma. The basic color of the ovary is pale pink, with no visible sign of pigmentation (Fig 4B).

## Genetic diversity among sweet fig banana populations

A total of 8,457 SNPs were obtained from Genotyping by sequencing, of which 86% were aligned with the reference genome of DH-Pahang V2. The proportion of SNP markers with

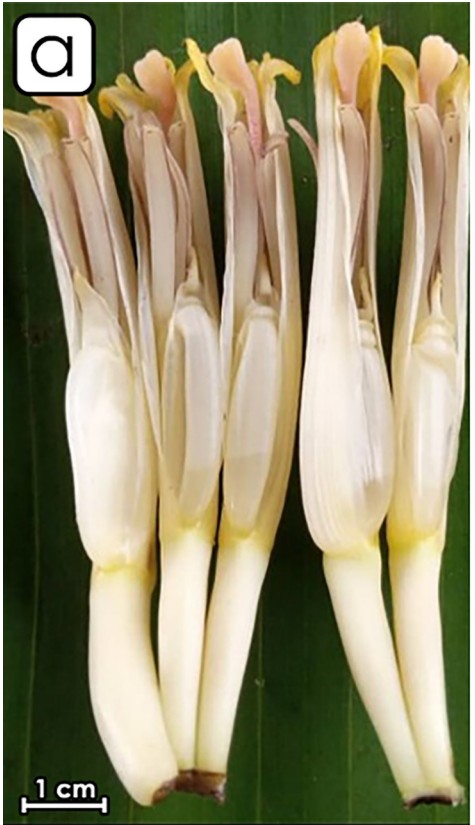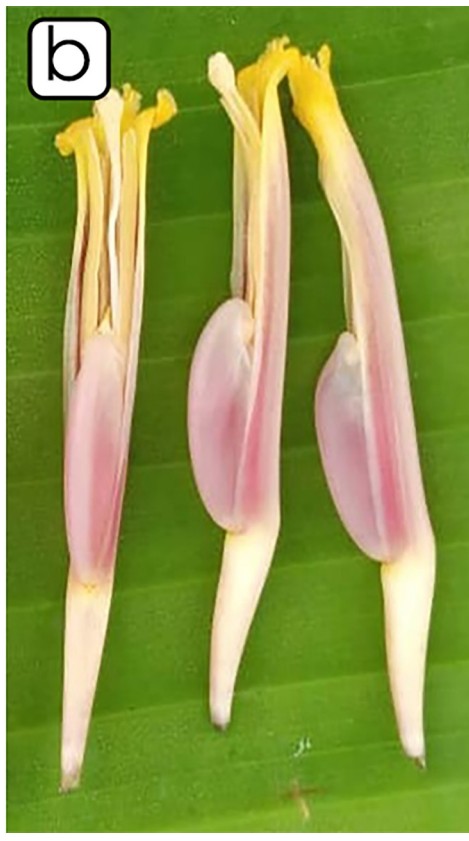

**Fig 4. Morphological aspect of the male flower of the two phenotypes of sweet fig banana.** (a) Major phenotype. (b) Minor phenotype.

missing data was 9.9%. The filtering of the markers resulted in a total of 1,992 polymorphic SNPs. The average PIC of these markers was 0.001 whereas the MAF was 0.05. Of the different types of polymorphism, transitions (53.8%) were more frequent than transversions (46.2%), with a transition/transversion rate of 1.16. The proportions of C/T and A/G transitions were similar (27.1% and 26.7%, respectively) as were those of A/T, G/T, and A/C transversions (14.2%, 12.7%, 12.6%, respectively).

The mean observed heterozygosity (Ho) and expected heterozygosity (He) of the overall sweet fig banana population were estimated at 0.0028 and 0.0162 respectively. The mean value of total gene diversity (Ht) was 0.0016. The average inbreeding coefficient (Fis) obtained from the total population was 0.82. The genetic diversity indices (Ho, He, Fis and Rs) varied across the eight sweet fig banana population (Table 1).

Overall, low levels of observed and expected heterozygosity were obtained, with Ho ranging from 0.001 (populations from the Departments of Atlantic, Mono and Ouémé-Plateau) to 0.004 (populations from the Department of Couffo) while expected heterozygosity (He) ranging from 0.003 (populations from the Departments of Mono and Ouémé-Plateau) to 0.054 (DOL population). Likewise, the inbreeding coefficients (Fis) were positive in all populations and ranged from 0.294 (LOA population) to 0.930 (DOL population). On average, the lowest allelic richness (Rs = 1.031) was observed in the SAD population against the highest value (Rs = 1.501) observed in the DOL population.

**Table 1. Statistics of genetic parameters among the eight sweet fig banana populations.**

| Pop | Department | N | Ho | He | Fis | Rs |
|------|------------|----|------|------|------|------|
| ZTA | Atlantique | 52 | 0.001 | 0.005 | 0.633 | 1.051 |
| LOA | Mono | 35 | 0.001 | 0.003 | 0.294 | 1.033 |
| GPO | Mono | 27 | 0.002 | 0.006 | 0.572 | 1.060 |
| DOL | Couffo | 27 | 0.004 | 0.054 | 0.930 | 1.501 |
| ZOO | Zou | 44 | 0.002 | 0.005 | 0.555 | 1.058 |
| SAD | Ouémé-Plateau | 18 | 0.001 | 0.003 | 0.382 | 1.031 |
| BBG | Collines-Donga | 57 | 0.003 | 0.033 | 0.926 | 1.352 |
| PST | Collines-Borgou | 13 | 0.003 | 0.011 | 0.605 | 1.086 |

Pop: populations; Dep: Department; N: Number of individuals; Ho: Observed heterozygosity; He: Expected heterozygosity; Fis: inbreeding coefficient, Rs: Allelic richness.

Population code: GPO: Grand-Popo; LOA: Lokossa, Athiémè; DOL: Dogbo, Lalo; ZTA: Zè, Tori-Bossito, Allada; SAD: Adjohoun, Sakété; ZOO: Zogbodomey, Ouinhi; BBG: Bantè; Bassila, Glazoué; PST: Parakou; Savè, Tchaourou.

## Population differentiation and structure

The overall differentiation index in the sweet fig banana germplasm was quasi-null (Fstg = -0.003); the pairwise Fstp values also reflected a total absence of differentiation between many pairs of populations (Table 2) (e.g. GPO from the sandy coastal plain of the department of Mono and SAD from the ferralitic soil plateau in the department of Ouémé and Plateau); the most differentiated populations are LOA and PST with Fstp = 0.03. Twelve out of the twenty-eight pairwise Fstp values were equal to 0. Meantime, the pairwise Fstp values of the remaining populations had non-zero Fstp values that ranged from 0.0123 to 0.0342.

The results of AMOVA revealed that 83.8% (Phi = 0.8387, $p < 0.001$) of the total molecular variance resulted from the genetic variation among samples within populations, whereas 16.1% (Phi = 0.8388; $p < 0.001$) was due to differences within samples (Table 3). The contribution of the occurrence region to molecular variance was quasi-null (0.85%, Phi = 0.0085; $p = 0.086$).

STRUCTURE analysis revealed a peak of deltaK at K = 3 suggesting the existence of three clusters in the germplasm (Fig 5A).

The three groups labeled in red (G1 with two accessions representing 0.7% of the overall population), blue (G2 with 258 accessions representing 94.5%), and green (G3 with 13 accessions representing 4.7% of the population) (Fig 5B).

**Table 2. Matrix of the pairwise Fstp value between the eight populations of sweet fig banana.**

|       | BBG | DOL | GPO | LOA | PST | SAD | ZOO | ZTA |
|-------|--------|--------|--------|--------|--------|------|--------|------|
| BBG | - |        |        |        |        |      |        |      |
| DOL | 0 | - |        |        |        |      |        |      |
| GPO | 0 | 0.0027 | - |        |        |      |        |      |
| LOA | 0 | 0.0125 | 0.0095 | - |        |      |        |      |
| PST | 0 | 0 | 0.0051 | 0.0342 | - |      |        |      |
| SAD | 0 | 0 | 0 | 0.0017 | 0.0077 | - |        |      |
| ZOO | 0 | 0.0156 | 0.0024 | 0 | 0.0123 | 0 | - |      |
| ZTA | 0.0028 | 0.0226 | 0.0072 | 0.0004 | 0.0129 | 0 | 0.0005 | - |

Population code: GPO: Grand-Popo; LOA: Lokossa, Athiémè; DOL: Dogbo, Lalo; ZTA: Zè, Tori-Bossito, Allada; SAD: Adjohoun, Sakété; ZOO: Zogbodomey, Ouinhi; BBG: Bantè; Bassila, Glazoué; PST: Parakou; Savè, Tchaourou.

**Table 3. Analysis of Molecular Variance (AMOVA) among and within the eight sweet fig banana populations.**

| Variation sources | Df | SS | MS | Sigma | %V | Phi | p-value |
|---|---|---|---|---|---|---|---|
| Between regions | 1 | 46.772 | 46.772 | 0.141 | 0.8 | 0.0085 | 0.086 |
| Between populations within the region | 7 | 164.718 | 23.531 | -0.133 | -0.8 | -0.0080 | 0.331 |
| Between samples within the populations | 264 | 8087.267 | 30.633 | 13.973 | 83.8 | 0.8387 | 0.001 |
| Between individual Within samples | 273 | 733.5 | 2.686 | 2.686 | 16.1 | 0.8388 | 0.001 |
| Total | 545 | 9032.258 | 16.572 | 16.668 | 100 | | |

Df: Degree of freedom; SS: Square Sum; MS: Mean Square; Sigma: Coefficient of Variance

%V: Percent of the total variance explained by each source of variance; Phi: Population differentiation statistics and p-value based on 999 permutations test.

The membership coefficient Q = 1 for G1; it varied from 0.931 to 1 for G2, and ranged from 0.645 to 0.897 for G3. Subgroup G1 accessions were genetically different from those of G2. Furthermore, based on the results of the STRUCTURE analysis, the accessions with a membership coefficient higher than 0.90 were considered to be pure. In contrast, accessions with a membership coefficient lower than 0.90 was considered admixture one. Therefore, the red and blue groups (95.2% of total accessions) consisted of pure accessions. Accessions in the green group were admixed.

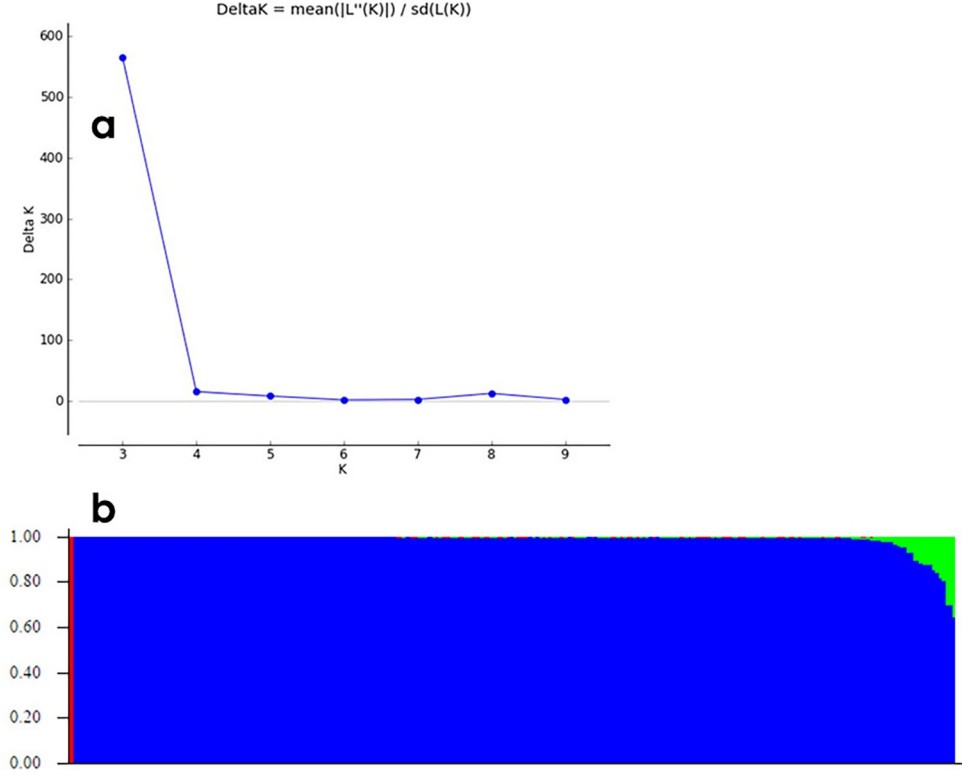

**Fig 5. Population structure using 1992 SNPs in STRUCTURE software.** (a) DeltaK for different number of sub-populations or genetic groups. (b) Plot for different genetic groups at K = 3, each color representing one genetic group. Red color for the first genetic group (G1), blue color for the second genetic group (G2) and green color for the third genetic group (G3).

## Cluster analysis

Cluster analysis dispatched the set of accessions to three groups, with the highest number (268) in cluster 2 (Fig 6).

This result provided by the neighbor-joining method matched with that obtained from the structure analysis. However, some discrepancies prevailed, as the group G3 in Structure

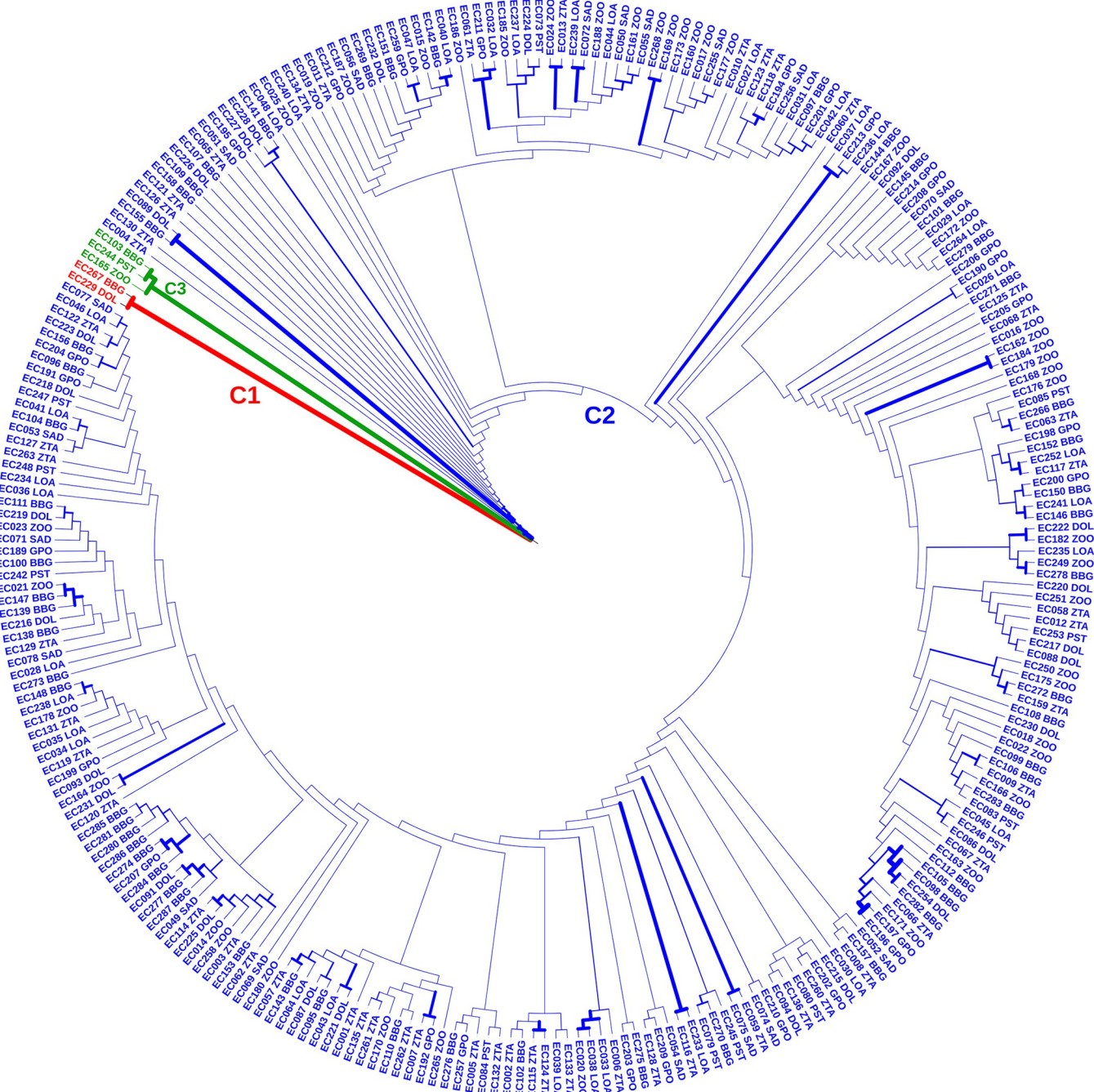

**Fig 6. Phylogenetic tree constructed with neighbor-joining (NJ) for 273 sweet fig banana accessions from Benin.** The topology revealed three genetic groups (C1: red; C2: blue and C3: green). The dendrogram was drawn using MEGA 7.

consisted of thirteen individuals instead of three as revealed by the neighbor-joining tree. Three accessions with a membership coefficient between 0.6 and 0.7 were categorized as admixtures cluster (C3) while those with a score higher than 0.8 were assigned to Cluster C2. Cluster C1 consisted of two accessions, EC229 and EC267, coming from two different climatic zones. Cluster C2 was the most complex, with 98.1% accessions.

Furthermore, STRUCTURE and Neighbor-joining results demonstrated that cluster 1 individuals were genetically different from those of cluster 2. This grouping clarified the phenotypic variations between the clusters 1 and 2 genotypes. Moreover, no morphological variation was noticed among the individuals of cluster 2 and 3.

Individuals in Cluster 2 and 3 exhibit the traits of the common sweet fig banana phenotype, with a light green false stem marked by large brown blotches. The bunch fruits are shaped like slightly compact truncated cone on downy peduncle. The fruit are stocky and curved, with male flower compound tepals that are cream-colored. Cluster 1 displays morphological traits of the minor phenotype with green-yellow pseudo stems marked by slight brown blotches, a cylindrical compact bunch on the hairless peduncle, long fruits, and pink-purple male flower compound tepals.

## Discussion

This research addressed the need for sufficient knowledge of the genetic diversity and population structure of sweet fig banana landraces in Benin, an economically valuable yet low-yielding crop threatened by biotic and abiotic stress. Understanding the genetic variation of crops facilitates decision-making about their conservation approach and can also help identify promising genotypes or alleles for crop improvement [58, 59]. In this study, we assessed the genetic variation of sweet fig banana germplasm. Our findings exposed the low genetic diversity of sweet fig banana germplasm in Benin and the absence of genetic structuring of the populations according to geographical occurrence.

### Genetic diversity in sweet fig banana germplasm

SNP data and genetic diversity parameters pointed out the near-uniformity of the germplasm of sweet fig banana landraces collected in the Guineo-Congolian and Sudano-Guinean regions of Benin. A total of 8,457 raw SNPs were yielded from the sweet fig banana genotypes. Prior investigation on *M. acuminata* ssp. *malaccensis* (PT-BA-00267) (9,968), a doubled haploid plant cv (DH-Pahang) yielded 9,968 SNPs [9, 47, 48]. The nucleotides composition was higher in transition substitutions than transversions, which was consistent with previous studies involving *Musa* spp [25, 60] and other plants species such as *Capsicum* spp. [61], *Cicer arietinum* L. [62], *Phaseolus vulgaris* L. [63]. This phenomenon, "known as transition bias," was attributed to a general property of DNA sequence evolution which expressed a better tolerance of DNA to transitional mutation than transversions during natural selection [64, 65]. In contrast, the mutations were observed mostly due to transversion substitution rather than transition [66]. In our study, GC transversion content was low, suggesting that the initial genetic diversity of SNPs in sweet fig bananas is highly conserved or has been enhanced over time. GC transversion content has an important influence on genome evolution, and its poor content indicates a low rate of mutation and recombination [67].

The mean values of observed heterozygosity (Ho) and expected heterozygosity (He) and the total gene diversity in sweet fig bananas were very low. The observed heterozygosity was lower than the expected heterozygosity for all populations, suggesting high levels of inbreeding. In connection, the high average inbreeding coefficient (Fis) suggested an excess of homozygotes and a narrow genetic basis in the sweet fig banana germplasm of Benin. A similar result was

observed by Kitavi et al. [68] and Němečková et al. [69] who reported that the triploids (AAA and AAB) cultivated East African Highland Banana (EAHB) subgroup was genetically uniform. Likewise, the edible East African diploid banana (AA) varieties were genetically homogenous although they exhibited high phenotypic variation and adaptation to diverse ecological zones. The diploid and triploid East African banana varieties belong to the same genetic complex [20].

A narrow genetic diversity can be due to putative severe genetic bottlenecks event that cultivated plants underwent during the initial domestication [70, 71]. However, this hypothesis contrasts with the high genetic diversity often reported for most *Musa acuminata* subspecies cultivars [17, 72, 73] and other *Musa* subspecies [74–76]. Indeed, the evolutionary process from wild to cultivated banana in *Musa acuminata*, deviates substantially from the general scenario in domestication which is known to reduce genetic diversity [77]. The current diversity in cultivated bananas most likely arose from multiple hybridization events between species and subspecies following the circulation of bananas pre-domesticated in multiple gene pools [8, 22]. According to the prevailing history, banana was domesticated from seven to nine wild species *Musa acuminata* (A genome). Through migration and seed dispersal in Southeast Asia and Western Melanesia, humans have brought plants of geographically isolated wild seminiferous subspecies of *M. acuminata* in contact, promoting natural intraspecific hybridization [3, 8, 23]. The crossings resulted in diploid (AA) hybrids with reduced fertility, which associated with human selection of seedless pulp led to parthenocarpic and edible fruits [8]. Other hybridization within *M acuminata* or with *M. Balbisiana* (B genome) species, sometimes involving diploid gamete formation, resulted in large base of the current diploid and triploid banana cultivars diversity [8, 78, 79]. This diversity includes bananas of various genomic configurations: AA, AAA, AB, AAB, and ABB [8, 80].

The accumulation of somatic mutation generates diversity, that can exclusively translate to a clonally derived landrace [22]. The heterozygosity deficiency in our study implies that not enough time has passed for the accumulation of spontaneous mutations through vegetative propagation, which could lead to significant differences between sweet fig banana populations. However, despite the low genetic divergence and a considerable degree of inbreeding in sweet fig banana populations, there was somewhat variation in the level of diversity among them. The most genetically diverse population was the DOL (Department of Couffo), followed by BBG (Centre-East Department) and PST populations (Centre-West). The high allelic richness in DOL and BBG populations suggested that these populations had the best long-term potential for adaptability and persistence in changing or stressful agroecological conditions.

## Genetic differentiation and population structure in sweet fig banana

According to Wright [51], Fst values below 0.05 imply little genetic difference. Our finding reveals poor genetic differentiation in the overall germplasm and among sweet fig banana populations. The low genetic differentiation among current populations of sweet fig banana can be explained by their sharing of ancestral common parent raised from historical exchange of genotypes for cultivation. In fact, the advent of gene flow in edible banana cultivars is no longer possible due to their parthenocarpy and sterility. Thereby, the occurrence of new cultivated banana varieties is limited to clonal fixation of mutations and rare spontaneous sexual events [81, 82].

The analysis of molecular variance showed that most of the genetic variation reside in the genetic variation within populations (83.8%) and did not depend at all on their geographical separation in the regions. Therefore, a higher level of genetic variation is observed within populations and a low genetic variation among populations. This pattern of genetic variation is

congruent with the findings reported from studies on the genetic diversity in *Musa acuminata* wild cultivars [83, 84] and outbreeding species [85, 86]. Huang et al. [87] demonstrated that purely clonal plants can conserve high genetic diversity within population like outbreeding plants and also preserve microgeographic genetic diversity through accumulation of somatic mutation. From this, we assumed that the level of genetic variation within populations may be due to mutation that occurred and fixed in some geographical populations and not yet spread to other populations.

Moreover, there was no significant genetic differentiation among sweet fig banana populations in the Guinean and Sudano-Guinean regions, probably because of their proximity and the extensive exchange of suckers among different growing communities. However, three clusters were identified in sweet fig banana germplasm. Almost all accessions (94.5%) were clustered in subgroup G2, which shows that most germplasm accessions were genetically uniform, regardless of their geographic and agroecological origins. The neighbor-joining phylogeny analysis split sweet fig banana genotypes into three lineages though the genotypes were mixed up in all clusters. A similar trend was observed in the phylogenetic study of Mulberry species [88] and *Musa* cultivars [89]. According to previous reports, the lack of a clear relationship between genotypes and their geographical origins is due to introduction followed by the naturalization of genotypes in areas away from their initial origin [88, 89]. Cluster C2 contained the majority (98.1%) of collected accessions that share the same genetic background. Furthermore, the individuals of Cluster 1 (0.7%) were genetically and morphologically different from the two other clusters. Cluster 3 (1.1%) is an admixture and did not show any phenotypic difference with the Cluster 1 individuals. Unlike individuals in Cluster 2 and 3 which exhibited almost all of the characteristics of *M. acuminata*, individuals in Cluster 1 exhibit a mixture of traits of *M. acuminata* and *M. balbisina* [6, 12, 90].

With reference to our findings, further investigations including flow cytometry ploidy analysis and morpho-taxonomic descriptors evaluation, will enable the confirmation of the genomic group of both accessions for accurate classification and conservation for use.

## Implications for conservation and breeding of sweet fig banana

The sterile and parthenocarpy nature of bananas, with the vegetative mode of propagation coupled with its genetic uniformity, make them particularly vulnerable to diseases and abiotic stress [68, 81]. The low genetic diversity may compromise the crop's ability to adapt to changing environmental conditions, making them more prone to extinction [91, 92].

Consequently, it is crucial to define effective conservation strategies and search for more variability that can be promoted for further cultivars development.

We suggest on-farm conservation of sweet fig banana accessions with other banana cultivars. This strategy allows for managing climatic risks and strengthens the resilience to pest and disease outbreaks [21]. The ex-situ conservation of sweet fig bananas should also be undertaken as an urgent matter through in vitro and in-vivo collections.

These findings can also help future genomic association studies in order to identify candidate genes associated with yield and other economically important traits.

The "Reconstructive breeding" approach developed by the French agricultural research and International Cooperation Organization (CIRAD) could be applied to produce improved hybrids of sweet fig banana cultivars. It consists initially of making a precise choice of the most relevant diploid parents among the probable ancestor of cultivars (Gros Michel or Cavendish subgroup), to develop a new improved triploid. This is followed by the assessment of the ability of diploid parents (of which one is a gamete donor) to combine with one other during hybridization and transfer the desired traits to offspring. This approach associates the favorable traits

brought by both parents to maximize the heterozygosity and heterosis in the new sterile trip-loid progenies [93]. In the case of sweet fig banana breeding establishment, the variety 'Cal-cutta 4' (*Musa acuminata*) can be used as the second parent. Indeed, 'Calcutta 4' exhibits an interesting source of resistance to the Sigatoka complex, yellow Sigatoka, Fusarium wilt, banana weevil, and burrowing nematodes [94]. Furthermore, considering the time consuming of the conventional breeding approach as well as the low yield of the cultivar 'Sotoumon' and stressful climatic conditions, it is necessary to complement classic breeding with biotechnology.

Other chemical and genomic tools, such as induced mutagenesis [95, 96], genomic selection [97–99], and CRISPR-Cas9 [100–102] can be used to develop a new cultivar with diverse traits of interest. The approaches mentioned above could be applied or combined to develop a high-yielding and climate-smart sweet fig banana variety with multiple and durable resistance to meet abiotic stresses (extreme temperature, drought, wind damage through dwarf plant size and strong root system to avoid), and biotic stresses (weevils, Sigatoka, Fusarium wilt, Banana Bunchy Top).

## Conclusion

The molecular characterization of sweet fig banana landraces from different regions of Benin based on SNPs markers shed light into a high genetic uniformity in germplasm and a little genetic differentiation among populations. Two phenotypes and three genetic groups were revealed for Benin. The population structure did not exhibit a clear relationship between sweet fig banana genotypes and their geographical/agroecological origins; most of accession (98%) were included in a unique major cluster. These results indicated that the crop's ability to face biotic and abiotic stress and the issue of climate change is seriously jeopardized. This situation calls for the necessity to deploy on farms and ex-situ germplasm conservation and genetic broadening of the germplasm. Our findings, the first in their kinds on the sweet fig banana, constitute a useful source of genetic information relevant for future genome-wide association studies and genomics selection breeding.

## Supporting information

**S1 Table. Geographical situation of the eight sweet fig banana populations.** This is a word file presenting the coordinates ranges of the studied banana populations.
(DOCX)

**S2 Table. Full list of single nucleotide polymorphism markers and associated metadata used in the molecular analysis.** This is a CSV formatted table.
(CSV)

## Acknowledgments

The authors would like to thank SEQART Africa for its support in the genotyping of acces-sions and local farmers for their consent to provide the plant materials used in this study.

## Author Contributions

**Conceptualization:** Dènoumi B. E. Capo-Chichi, Enoch G. Achigan-Dako.

**Formal analysis:** Dènoumi B. E. Capo-Chichi, Dèdéou A. Tchokponhoué.

**Investigation:** Dènoumi B. E. Capo-Chichi.

**Methodology:** Dènoumi B. E. Capo-Chichi, Enoch G. Achigan-Dako.

**Project administration:** Dènoumi B. E. Capo-Chichi, Enoch G. Achigan-Dako.

**Supervision:** Enoch G. Achigan-Dako.

**Validation:** Dèdéou A. Tchokponhoué, Dêêdi E. O. Sogbohossou.

**Visualization:** Dènoumi B. E. Capo-Chichi, Dèdéou A. Tchokponhoué.

**Writing – original draft:** Dènoumi B. E. Capo-Chichi.

**Writing – review & editing:** Dèdéou A. Tchokponhoué, Dêêdi E. O. Sogbohossou, Enoch G. Achigan-Dako.

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
