## [Decision Letter · Decision Letter 0]

28 Sep 2023

PONE-D-23-29467Narrow genetic diversity in germplasm from the Guinean and Sudano-Guinean zones in Benin indicates the need to broaden the genetic base of sweet fig banana (Musa acuminata cv Sotoumon)PLOS ONE

Dear Dr. CAPO-CHICHI,

Thank you for submitting your manuscript to PLOS ONE. After careful consideration, we feel that it has merit but does not fully meet PLOS ONE’s publication criteria as it currently stands. Therefore, we invite you to submit a revised version of the manuscript that addresses the points raised during the review process.

We look forward to receiving your revised manuscript.

Kind regards,

Valentine Otang Ntui, Ph.D

Academic Editor

PLOS ONE

Journal Requirements:

2. Thank you for stating the following in your Competing Interests section: "The authors declare that the research was conducted in the absence of any commercial or financial relationships that could be construed as a potential conflict of interest".

Reviewers' comments:

Reviewer's Responses to Questions

**Comments to the Author**

1. Is the manuscript technically sound, and do the data support the conclusions?

Reviewer #1: Yes

Reviewer #2: Yes

2. Has the statistical analysis been performed appropriately and rigorously? 

Reviewer #1: Yes

Reviewer #2: Yes

3. Have the authors made all data underlying the findings in their manuscript fully available?

Reviewer #1: Yes

Reviewer #2: Yes

4. Is the manuscript presented in an intelligible fashion and written in standard English?

Reviewer #1: Yes

Reviewer #2: Yes

5. Review Comments to the Author

Reviewer #1: The authors assessed the genetic diversity of an important banana variety in Benin, which is potentially crucial for variety conservation and improvement. The study is original; the data is adequate, well-presented and discussed. I believe these findings deserve dissemination. Thus, I recommend publishing this work after rectifying the minor issues listed below.

Line 59: Change ‘were not evaluated’ to ‘have not been evaluated’.

Line 104: Change ‘young leaves samples’ to ‘young leaf samples’.

Line 106: Put ‘Nairobi, Kenya’ in brackets after SEQART AFRICA.

Line 108: Change ‘controlled’ to ‘confirmed’.

Line 128 and Line 132: Check the empty brackets.

Line 137: Replace ‘carried out’ with ‘performed’ for conciseness.

Line 298: Remove the comma after ‘According..’.

Line 351: Change ‘others’ to ‘other’.

Line 354: Change ‘this’ to ‘these’.

Line 359: Change ‘follow’ to ‘followed’.

Reviewer #2: Generally, this paper is well-written, clear and fills a knowledge gap in the genetic diversity of the Musa species. The work was well done, with a well-thought-out research plan, data analysis and presentation.

However, the paper has a few technical and language pitfalls that need to be corrected, as summarized below:

1. Check the entire article for grammatical errors, including pronoun disagreement, misplaced modifiers, missing/ wrong punctuation, lack of subject/ verb agreement, inconsistent/ wrong spacing of words, incomplete comparisons/ descriptions, use of wrong prepositions, and compound sentences among others. Here are some of the issues noted.

Line 59 – ‘Have not been’ instead of ‘were not’.

Line 70, use the singular form ‘rainfall’ instead of ‘rainfalls’.

Line 104 – use the singular form ‘leaf’.

Line 115 – The plural verb ‘were’ agrees with the plural noun ‘SNPs’. Correct this.

Line 250 – Correct the spellings for ‘with’.

Line 306 – Delete ‘be’.

Line 328 – Change others to other.

Line 351 – Change others to other.

Line 354 – Change ‘this’ to ‘these’.

Line 374 – Change lights to the singular form light.

Line 376 – Delete the space between origins and the semi-colon.

2. Check your citations and references for uniformity and that they abide by the journal recommendations. E.g., Line 337 – Put citation numbers 88 and 89 under one parenthesis.

3. Figures, Fig 3 caption is unconcise and confusing to the reader, with unnecessary repetitions in the sub-heading. Change it to “Morphology of the male bud in the two phenotypes of sweet fig banana”. (a) major phenotype; (b) minor phenotype. Fig 4 caption: As stated above, there is no need to re-write ‘of sweet fig banana’ in A and B since it's written in the main caption.

4. Avoid unnecessary repetition e.g., Line 179 – It is already clear to the reader that you were working on of sweet fig banana so there is no need to write that repeatedly. Delete the areas in the article you have over-used the phrase.

6. PLOS authors have the option to publish the peer review history of their article (what does this mean?). If published, this will include your full peer review and any attached files.

Reviewer #1: No

Reviewer #2: **Yes: **Easter D. Syombua

---

## [Author Response · Author response to Decision Letter 0]

19 Oct 2023

Dènoumi Béatrice Elodie CAPO-CHICHI

Genetics, Biotechnology and Seed Science 

Unit (GBioS), University of Abomey-Calavi, 

01 BP 526, Abomey-Calavi,

Republic of Benin

October 19, 2023

Subject: Rebuttal Letter

Prof. Emily Chenette

Cambridge, England, United Kingdom,

Chief Editor of Journal PLoS ONE.

Dear Prof. Emily Chenette,

The authors thank the academic editor and the two reviewers for their useful comments about our manuscript entitled “Narrow Genetic Diversity in Germplasm from the Guinean and Sudano-Guinean Zones in Benin indicates the need to broaden the genetic base of Sweet Fig Banana (Musa acuminata cv Sotoumon).” All the comments and suggestions were carefully addressed, and changes made in the revised manuscript were tracked. We have included the revised manuscript with tracked changes and an unmarked version of the revised paper without tracked changes when resubmitting the new version of the manuscript. 

The study was unfunded, and the author(s) received no specific funding for this work. We want to keep our financial disclosure. 

Moreover, the laboratory protocol used in the current study for genotyping using Diversity Arrays Technology (DArT) markers is the in-house protocol of SEQART AFRICA (Nairobi, Kenya), which is described in the materials and method. Consequently, we cannot make a deposit of or publish this laboratory protocol. 

Further, below is the point-by-point response to the comments and suggestions by the editor and the reviewers.

Sincerely yours,

Dènoumi CAPO-CHICHI

 

Manuscript number: PONE-D-23-29467

Manuscript title: Narrow genetic diversity in germplasm from the Guinean and Sudano-Guinean zones in Benin indicates the need to broaden the genetic base of sweet fig banana (Musa acuminata cv Sotoumon)

Point-by-point response to the academic editor and reviewers’ comments

1. Academic editor

Comment 1. Response to comment 1. This study was unfunded, therefore the author(s) received no specific funding for this work. We want to keep our financial disclosure.

Comment 2. Guidelines for resubmitting your figure files are available below the reviewer comments at the end of this letter.

Response to comment 2. We have considered your recommendation and uploaded the figure files to the digital diagnostic tool Preflight Analysis and Conversion Engine (PACE), during the submission review. We can assure you that the figures meet PLOS requirements. 

Comment 3. If applicable, we recommend that you deposit your laboratory protocols in protocols.io to enhance the reproducibility of your results. Protocols.io assigns your protocol its own identifier (DOI) so that it can be cited independently in the future. For instructions see: https://journals.plos.org/plosone/s/submission-guidelines#loc-laboratory-protocols. Additionally, PLOS ONE offers an option for publishing peer-reviewed Lab Protocol articles, which describe protocols hosted on protocols.io. Read more information on sharing protocols at https://plos.org/protocols?utm_medium=editorialemail&utm_source=authorletters&utm_campaign=protocols.

Response to comment 3. We thank the Editor for this insight, but the laboratory protocol used in the current study for genotyping using Diversity Arrays Technology (DArT) markers is the in-house protocol of SEQART AFRICA (Nairobi, Kenya), which we have described in the materials and method. Therefore, we cannot make a deposit of or publish the laboratory protocol.

2. Journal Requirements

Comment 1. Please ensure that your manuscript meets PLOS ONE's style requirements, including those for file naming. The PLOS ONE style templates can be found at 

Response to comment 1. We have downloaded the templates and revised the manuscript bases on PLOS ONES’s style requirements, including those for file naming.

Comment 2. Thank you for stating the following in your Competing Interests section: "The authors declare that the research was conducted in the absence of any commercial or financial relationships that could be construed as a potential conflict of interest".

Response to comment 2

We have now specified the following in the "Competing interests" section: "The authors declare that the research was conducted in the absence of any commercial or financial relationship that could be interpreted as a potential conflict of interest".

Comment 3. Please complete your Competing Interests on the online submission form to state any Competing Interests. If you have no competing interests, please state "The authors have declared that no competing interests exist.", as detailed online in our guide for authors at http://journals.plos.org/plosone/s/submit-now

. This information should be included in your cover letter; we will change the online submission form on your behalf.

Response to comment 3. Thank you for your recommendations. 

 We have completed the Competing Interests section on the online submission form to indicate the absence of any competing interests. We have stated: "The authors have declared that there are no competing interests."

Likewise, information about competing interests is now included in our cover letter. The statement reads: "The authors have declared that no competing interests exist."

Comment 4. We note that you have indicated that data from this study are available upon request. PLOS only allows data to be available upon request if there are legal or ethical restrictions on sharing data publicly. For more information on unacceptable data access restrictions, please see http://journals.plos.org/plosone/s/data-availability#loc-unacceptable-data-access-restrictions. 

 b) If there are no restrictions, please upload the minimal anonymized data set necessary to replicate your study findings as either Supporting Information files or to a stable, public repository and provide us with the relevant URLs, DOIs, or accession numbers. For a list of acceptable repositories, please see http://journals.plos.org/plosone/s/data-availability#loc-recommended-repositories. We will update your Data Availability statement on your behalf to reflect the information you provide.

Response to comment 3. There are no ethical or legal restrictions on sharing a de-identified data set. All relevant data necessary to replicate our study findings have now been uploaded as supporting information files (S1 and S2 Tables). 

Comment 4. Please include your full ethics statement in the ‘Methods’ section of your manuscript file. In your statement, please include the full name of the IRB or ethics committee who approved or waived your study, as well as whether or not you obtained informed written or verbal consent. If consent was waived for your study, please include this information in your statement as well.

Response to comment 4. We have included the ethics statement in our manuscript in the “Materials and Methods” section entitled "National and local approval" from lines 115 to 124.

This study is part of a doctorate thesis whose protocol was approved by the Academic Committee of the Faculty of Agronomic Sciences (FSA), University of Abomey-Calavi (UAC), Abomey-Calavi, Republic of Benin. The plant materials were collected following the National and International Code of Conduct for plant germplasm collection. Before collection, we presented the study's aim and methodology to the local authorities and sweet fig banana plant owners of each village investigated. Then, we required their informed verbal consent to collect the seedlings. We obtained a verbal agreement from each local authority and donor before proceeding to suckers collection. All plant materials are conserved at the Genetics, Biotechnology and Seed Sciences Unit (GBioS), Laboratory of Crop Production, Physiology, and Plant Breeding, Faculty of Agronomic Sciences, University of Abomey-Calavi (Republic of Benin).

Response to comment 5. We note that Figure 1 in your submission contains [map/satellite] images which may be copyrighted. All PLOS content is published under the Creative Commons Attribution License (CC BY 4.0), which means that the manuscript, images, and Supporting Information files will be freely available online, and any third party is permitted to access, download, copy, distribute, and use these materials in any way, even commercially, with proper attribution. For these reasons, we cannot publish previously copyrighted maps or satellite images created using proprietary data, such as Google software (Google Maps, Street View, and Earth). For more information, see our copyright guidelines: http://journals.plos.org/plosone/s/licenses-and-copyright.

 1. You may seek permission from the original copyright holder of Figure 1 to publish the content specifically under the CC BY 4.0 license. We recommend that you contact the original copyright holder with the Content Permission Form (http://journals.plos.org/plosone/s/file?id=7c09/content-permission-form.pdf) and the following text: “I request permission for the open-access journal PLOS ONE to publish XXX under the Creative Commons Attribution License (CCAL) CC BY 4.0 (http://creativecommons.org/licenses/by/4.0/). Please be aware that this license allows unrestricted use and distribution, even commercially, by third parties. Please reply and provide explicit written permission to publish XXX under a CC BY license and complete the attached form.” Please upload the completed Content Permission Form or other proof of granted permissions as an ""Other"" file with your submission. In the figure caption of the copyrighted figure, please include the following text: “Reprinted from [ref] under a CC BY license, with permission from [name of publisher], original copyright [original copyright year].”

 2. If you are unable to obtain permission from the original copyright holder to publish these figures under the CC BY 4.0 license or if the copyright holder’s requirements are incompatible with the CC BY 4.0 license, please either i) remove the figure or ii) supply a replacement figure that complies with the CC BY 4.0 license. Please check copyright information on all replacement figures and update the figure caption with source information. If applicable, please specify in the figure caption text when a figure is similar but not identical to the original image and is therefore for illustrative purposes only. The following resources for replacing copyrighted map figures may be helpful: USGS National Map Viewer (public domain): http://viewer.nationalmap.gov/viewer/ The Gateway to Astronaut Photography of Earth (public domain):http://eol.jsc.nasa.gov/sseop/clickmap/

Response to comment 5. Thank you for your recommendation and guidance for Figure 1 to meet the free available online standards. After carefully reading your guidance, we supplied a replacement figure that complies with the CC BY 4.0 license. This new figure has been designed using a base map from the National Geographic Institute of Benin in French « Institut Géographique National du Bénin (IGN) ». IGN base map is in the public domain in Benin. We updated the Figure 1 caption by including the source information of the base map. (See lines 90 to 91, revised manuscript).

Comment 6. Please review your reference list to ensure that it is complete and correct. If you have cited papers that have been retracted, please include the rationale for doing so in the manuscript text or remove these references and replace them with relevant current references. Any changes to the reference list should be mentioned in the rebuttal letter that accompanies your revised manuscript. If you need to cite a retracted article, indicate the article’s retracted status in the References list and also include a citation and full reference for the retraction.

Response to comment 6. 

All references were double-checked; no retracted paper was cited.

3. Reviewer comments

#Reviewer 1

Overall comments to the Author

The authors assessed the genetic diversity of an important banana variety in Benin, which is potentially crucial for variety conservation and improvement. The study is original; the data is adequate, well-presented and discussed. I believe these findings deserve dissemination.

Response to reviewer 1’s overall comment

We thank the reviewer 1 for their positive feedback.

Reviewer 1 specific comments

Comment 1. Line 59: Change ‘were not evaluated’ to ‘have not been evaluated’.

Response to reviewer 1’s comment 1. We have replaced the terms ‘were not evaluated’ by ‘have not been evaluated’ (see line 69, revised manuscript). 

Comment 2. Line 104: Change ‘Young leaves samples’ to ‘young leaf samples’.

Response to reviewer 1’s comment 2. “Young leaves” has been replaced by “young leaf” (see line 127, revised manuscript).

Comment 3. Line 106: Put ‘Nairobi, Kenya’ in brackets after SEQART AFRICA.

Response to reviewer 1’s comment 3. (Nairobi, Kenya) is now in bracket. (see line 129, revised manuscript).

Comment 4. Line 108: Change ‘controlled’ to ‘confirmed’

Response to reviewer 1’s comment 4. ‘controlled’ now reads ‘confirmed’ (see line 131, revised manuscript)..

Comment 5. Line 128 and Line 132: Check the empty brackets. 

Response to reviewer 1’s comment 5. The so-called empty brackets have a coding meaning. Indeed, they are always preceded by a statistical function of the R environment [e.g., stamppFst ()]; the italicized empty brackets meant that there were arguments (not explicitly presented due to their length) that were passed to the function. The brackets have been maintained.

Comment 6. Line 137: Replace ‘carried out’ with ‘performed’ for conciseness.

Response to reviewer 1’s comment 6. ‘Carried out’ has been replaced by ‘performed’ (see line 161, revised manuscript). 

Comment 7. Line 298: Remove the comma after ‘According..’

Response to reviewer 1’s comment 7. Done (see line 324, revised manuscript).

Comment 8. Line 351: Change ‘others’ to ‘other’.

Response to reviewer 1’s comment 8. “Others” now reads “other” (see line 366, revised manuscript).

Comment 9. Line 354: Change ‘this’ to ‘these’.

Response to reviewer 1’s comment 9. “This” is now replaced by “these” (see line 381, revised manuscript). 

Comment 10. Line 359: Change ‘follow’ to ‘followed’.

Response to reviewer 1’s comment 10. ‘follow’ now reads ‘followed’ (see line 386, revised manuscript).

Reviewer #2

Overall comments to the Author

Generally, this paper is well-written, clear and fills a knowledge gap in the genetic diversity of the Musa species. The work was well done, with a well-thought-out research plan, data analysis and presentation. 

Response to reviewer 2’s overall comment

We thank the reviewer 2 for their positive feedback.

Reviewer 2 specific comments

Comment 1. However, the paper has a few technical and language pitfalls that need to be corrected, as summarized below: Check the entire article for grammatical errors, including pronoun disagreement, misplaced modifiers, missing/ wrong punctuation, lack of subject/ verb agreement, inconsistent/ wrong spacing of words, incomplete comparisons/ descriptions, use of wrong prepositions, and compound sentences among others. Here are some of the issues noted. Q1. Line 59 – ‘Have not been’ instead of ‘were not’. Q2. Line 70, use the singular form ‘rainfall’ instead of ‘rainfalls’. 

Q3. 104 – use the singular form ‘leaf. Q4. Line 115 – The plural verb ‘were’ agrees with the plural noun ‘SNPs’. Correct this. Q5. Line 250 – Correct the spellings for ‘with’. 

Q6. Line 306 – Delete ‘be’. Q7. Line 328 – Change others to other. 

Q8. Line 351 – Change others to other. Q9. Line 354 – Change ‘this’ to ‘these’. 

Q10. Line 374 – Change lights to the singular form light. 

Q11. Line 376 – Delete the space between origins and the semi-colon.

Response to reviewer 2's comment 1. We are grateful to the reviewer for their comments All recommended corrections were brought in the revised manuscript as follows:

Q1. Line 59 – ‘Have not been’ instead of ‘were not’.

A1. Done (see line 69, revised manuscript).

Q2. Line 70, use the singular form ‘rainfall’ instead of ‘rainfalls’. 

A2. Done (see line 80, revised manuscript).

Q3. Line 104 – use the singular form ‘leaf’.

A3. Done (see line 127, revised manuscript).

Q4. Line 115 – The plural verb ‘were’ agrees with the plural noun ‘SNPs’. Correct this. 

A4. Corrected Done (see line 139, revised manuscript).

Q5. Line 250 – Correct the spellings for ‘with’.

A 5. This is corrected Done (see line 276, revised manuscript).

Q6. Line 306 – Delete ‘be’.

A6. Done (see line 332, revised manuscript).

Q7. Line 328 – Change others to other.

A7. Done (see lines 354, revised manuscript).

Q8. Line 351 – Change others to other.

A8. Done (see line 365, revised manuscript).

Q9. Line 354 – Change ‘this’ to ‘these’.

A9. Done (see line 381, revised manuscript).

Q10. Line 374 – Change lights to the singular form light.

A10. Done (see line 402, revised manuscript)..

Q11.Line 376 – Delete the space between origins and the semi-colon. 

A.11. The space between origins and the semi-colon is now deleted Done (see line 404, revised manuscript).

Comment 2. Check your citations and references for uniformity and that they abide by the journal recommendations. E.g., Line 337 – Put citation numbers 88 and 89 under one parenthesis.

Response to reviewer 2’s comment 2. Citations have been checked and references aligned with the journal's recommendations. The citation numbers 88 and 89 have now been put in one parenthesis (see line 363, revised manuscript).

Comment 3. Figures, Fig 3 caption is unconcise and confusing to the reader, with unnecessary repetitions in the sub-heading. Change it to “Morphology of the male bud in the two phenotypes of sweet fig banana”. (a) major phenotype; (b) minor phenotype. Fig 4 caption: As stated above, there is no need to re-write ‘of sweet fig banana’ in A and B since it's written in the main caption. 

Response to reviewer 2’s comment 3

 Figures 2, 3 and 4 captions have been rephrased. (see lines 177 to 178 for Figure 2, lines 183 to 186 for Figure 3 and lines 193 to 194, revised manuscript).

Comment 4. Avoid unnecessary repetition e.g., Line 179 – It is already clear to the reader that you were working on of sweet fig banana so there is no need to write that repeatedly. Delete the areas in the article you have over-used the phrase.

Response to reviewer 2’s comment 4.

Thank you for this recommendation. These sentences were removed from the manuscript the lines (see lines 166, 204-205, 257 and 270, revised manuscript).

---

## [Editor Report · Decision Letter 1]

31 Oct 2023

Narrow genetic diversity in germplasm from the Guinean and Sudano-Guinean zones in Benin indicates the need to broaden the genetic base of sweet fig banana (Musa acuminata cv Sotoumon)

PONE-D-23-29467R1

Dear Dr. Capo-Chichi,

We’re pleased to inform you that your manuscript has been judged scientifically suitable for publication and will be formally accepted for publication once it meets all outstanding technical requirements.

Kind regards,

Valentine Otang Ntui, Ph.D

Academic Editor

PLOS ONE
---

## [Editor Report · Acceptance letter]

9 Nov 2023

PONE-D-23-29467R1 

Narrow genetic diversity in germplasm from the Guinean and Sudano-Guinean zones in Benin indicates the need to broaden the genetic base of sweet fig banana (*Musa acuminata* cv Sotoumon) 

Dear Dr. Capo-Chichi:

I'm pleased to inform you that your manuscript has been deemed suitable for publication in PLOS ONE. Congratulations! Your manuscript is now with our production department. 

Kind regards, 

on behalf of

Dr. Valentine Otang Ntui 

Academic Editor

PLOS ONE